# Decreased Epicardial *CTRP3* mRNA Levels in Patients with Type 2 Diabetes Mellitus and Coronary Artery Disease Undergoing Elective Cardiac Surgery: A Possible Association with Coronary Atherosclerosis

**DOI:** 10.3390/ijms23179988

**Published:** 2022-09-01

**Authors:** Zdenek Matloch, Milos Mraz, Barbora Judita Kasperova, Helena Kratochvilova, Petr Svoboda, Iveta Pleyerova, Katerina Reznickova, Sarah Norman, Daniel Hlavacek, Jakub Mahrik, Peter Ivak, Zdenka Lacinova, Ivan Netuka, Martin Haluzik

**Affiliations:** 1Department of Cardiothoracic Anaesthesia and Intensive Care, The James Cook University Hospital, Marton Road, Middlesbrough TS4 3BW, UK; 2First Faculty of Medicine, Charles University, Katerinska 32, 121 08 Prague, Czech Republic; 3Diabetes Centre, Institute for Clinical and Experimental Medicine, Videnska 1958/9, 140 21 Prague, Czech Republic; 4Institute of Medical Biochemistry and Laboratory Diagnostics, First Faculty of Medicine, Charles University and General University Hospital, U Nemocnice 499/2, 128 08 Prague, Czech Republic; 5Experimental Medicine Centre, Institute for Clinical and Experimental Medicine, Videnska 1958/9, 140 21 Prague, Czech Republic; 6Department of Biochemistry and Microbiology, University of Chemistry and Technology Prague, Technicka 5, 166 28 Prague, Czech Republic; 7Department of Anaesthesia, The James Cook University Hospital, Marton Road, Middlesbrough TS4 3BW, UK; 8Department of Cardiac Surgery, Institute for Clinical and Experimental Medicine, Videnska 1958/9, 140 21 Prague, Czech Republic; 9Department of Physiology, Third Faculty of Medicine, Charles University, Ruska 87, 100 00 Prague, Czech Republic

**Keywords:** adipokines, epicardial adipose tissue, coronary artery disease, diabetes mellitus, gene expression, CTRP3

## Abstract

(1) Background: C1q TNF-related protein 3 (CTRP3) is an adipokine with anti-inflammatory and cardioprotective properties. In our study, we explored changes in serum CTRP3 and its gene expression in epicardial (EAT) and subcutaneous (SAT) adipose tissue in patients with and without coronary artery disease (CAD) and type 2 diabetes mellitus (T2DM) undergoing elective cardiac surgery. (2) Methods: SAT, EAT, and blood samples were collected at the start and end of surgery from 34 patients: (i) 11 without CAD or T2DM, (ii) 14 with CAD and without T2DM, and (iii) 9 with both CAD and T2DM. mRNA levels of *CTRP3* were assessed by quantitative reverse transcription PCR. Circulating levels of CTRP3 and other factors were measured using ELISA and Luminex Multiplex commercial kits. (3) Results: Baseline plasma levels of TNF-α and IL6 did not differ among the groups and increased at the end of surgery. Baseline circulating levels of CTRP3 did not differ among the groups and decreased after surgery. In contrast, baseline *CTRP3* mRNA levels in EAT were significantly decreased in CAD/T2DM group, while no differences were found for *TNF-**α* and *IL6* gene expression. (4) Conclusions: Our data suggest that decreased EAT mRNA levels of *CTRP3* could contribute to higher risk of atherosclerosis in patients with CAD and T2DM.

## 1. Introduction

Cardiovascular diseases (CVD) are among the most common causes of morbidity and mortality in patients with obesity and type 2 diabetes mellitus (T2DM). These disorders are associated with other concomitant diseases, such as hyperlipidemia and arterial hypertension, leading to the development of atherosclerosis and coronary artery disease (CAD) [1,2]. Atherosclerosis is characterized by deposits of immune cells and lipids in the subendothelial space of arteries. CAD is a complex process not only involving intra-arterial accumulation of calcium and cholesterol, but also consisting of the inflammatory response driven by both cytokines and pro-inflammatory cells within vascular endothelium too [3,4]. Endothelial cells function as a semipermeable layer that separates the arterial wall from the intravascular components [5,6]. This barrier prevents platelet aggregation, regulates vascular tone, and enables fluid homeostasis. The endothelium produces vasoconstrictor and vasodilator molecules such as endothelin and nitric oxide, respectively. The disproportional production of these vasoactive substances may contribute to the loss of endothelial homeostatic function facilitating endothelial dysfunction [7,8]. This together with inflammatory response promote thrombosis and lead to adverse cardiovascular events [9]. A cross-link between CAD and the volume, quality, and overall layout of epicardial adipose tissue is well documented [10].

Epicardial fat is unique in its function and anatomical proximity to the heart; it affects myocardium metabolism and the development of CAD as well as other diseases [11,12,13]. In 2004, Wong et al. described a novel family of adipose tissue-derived cytokines referred to as complement C1q TNF-related proteins (CTRPs) [14]. C1q TNF-related protein 3 (CTRP3) (aka CORS26, cartducin, or cartonectin) is a unique member of this adipokine family. It is expressed and released by most types of human tissues (liver, heart, kidney, lung, cartilage, and bone) including adipose tissue. The CTRPs, a family of highly conserved adiponectin paralogs, are secreted molecules consisting of four different domains: an N-terminal signal peptide, a collagen-like domain, a short variable domain, and a C-terminal C1q-like globular domain [15]. CTRP3 is an adipokine that inhibits pro-inflammatory pathways in microcells and monocytes, exerting anti-apoptotic and cardioprotective effects during the development of CAD [16,17]. CTRP3 influences nuclear factor kappa B (NF-κB) and inhibitory toll-like receptors (TLRs) signaling pathways to reduce the secretion of pro-inflammatory adipocytokines, thus attenuating both obesity-related, chronic, systemic pro-inflammatory responses, and insulin resistance [18,19]. Additionally, CTRP3 enhances the activation of the PI3K/Akt/eNOS pathway, which inhibits endothelial dysfunction induced by oxidized low-density lipoproteins (ox-LDL), thus downregulating an expression of *TNF-α*, *CTRPs*, and *IL6*, postponing atherosclerosis progression [20].

As CTRP3 has documented anti-inflammatory and cardioprotective properties, we hypothesized that its epicardial adipose tissue gene expression or circulating levels may differ in patients with T2DM and CAD, compared to patients without those conditions thus contributing to accelerated coronary atherosclerosis. To this end, we explored changes of CTRP3 circulating levels and its mRNA levels in epicardial (EAT) and subcutaneous (SAT) adipose tissue in patients with or without CAD and T2DM undergoing elective cardiac surgery.

## 2. Results

### 2.1. Anthropometric and Biochemical Characteristics

At baseline, there was little difference between the three groups of patients in the presented anthropometric and biochemical characteristics. By definition, fasting plasma glucose and HbA1c levels were significantly higher in +CAD+T2DM group. No differences in the parameters of glucose control were found between -CAD-T2DM and +CAD-T2DM (Table 1).

### 2.2. Serum Concentrations of Selected Pro-Inflammatory Mediators and Adipokines

Serum concentrations of measured adipokines and cytokines are shown in Table 2. Baseline levels of CTRP3 as well as hsCRP, leptin, adiponektin, IL6, and TNF-α did not differ between the groups. As expected, cardiac surgery was associated with increased serum IL6 and TNF-α, while at the same time leading to a partially significant reduction of hsCRP, adiponectin, and leptin in some of the groups. In contrast, circulating CTRP3 levels were decreased consistently and significantly by surgery in all study groups.

### 2.3. Relative mRNA Quantification of TNF-α, CTRP3, and IL6 in EAT and SAT

In EAT, baseline *CTRP3* mRNA levels were significantly reduced in +CAD+T2DM relative to -CAD-T2DM subjects, while no difference was seen in baseline *TNF-α* or *IL6* gene expression between any of the groups. Surgery did not significantly affect *CTRP3* mRNA levels (albeit a non-significant increase was observed in all groups), while markedly increasing *IL6* and, to a lesser extent, also *TNF-α* gene expression (Figure 1; Appendix A).

In SAT, in contrast, *CTRP3* mRNA levels did not differ between groups either at baseline or after surgery, with surgery rather tending to slightly reduce *CTRP3* gene expression. Both *IL6* and *TNF-α* showed the same mRNA level pattern as in EAT with no difference at baseline and a consistent operation-induced increase throughout the groups (Figure 2; Appendix A).

## 3. Discussion

Adipose tissue covers about 80% of the heart’s surface and forms up to 20% of its weight [10,21]. Endocrine function of adipose tissue is markedly affected by the presence of T2DM and obesity, with decreased production of anti-inflammatory and increased production of pro-inflammatory factors relative to healthy subjects [22,23]. This endocrine dysfunction contributes to the development of cardiovascular and metabolic complications of obesity and T2DM [22,24]. Our study focused on the novel adipokine CTRP3 dynamics in serum and its mRNA quantification in both EAT and SAT in patients undergoing elective cardiac surgery. CTRP3 has been shown to improve glucose and lipid metabolism and to have anti-inflammatory and cardioprotective properties [25,26]. To the best of our knowledge, changes of CTRP3 in epicardial fat in patients with differential presence of coronary atherosclerosis has not been studied so far.

Our study shows that neither the presence of T2DM or CAD affected circulating CTRP3 levels or their change after cardiac surgery. However, the literature regarding CTRP3 and T2DM (and its associated conditions) is inconsistent. Previously, CTRP3 levels were described to be strongly affected by metabolic status, obesity, or changes in glucose metabolism. Circulating CTRP3 levels were increased by fasting [27] and the presence of pre-diabetes [28]. On the contrary, the presence of T2DM was accompanied by decreased CTRP3 levels [29,30,31,32]. Furthermore, several studies illustrated that CTRP3 levels were increased with treatment by metformin [33] or Glucagon-like peptide 1 receptor agonists [34], which indicates that its increase could potentially contribute to improved glucose control. Lack of differences in baseline CTRP3 concentrations between +T2DM and -T2DM groups in our study could be related to the smaller sample size as well as to the presence of multiple comorbidities including coronary artery disease that could have affected its circulating levels.

In all groups, CTRP3 concentrations significantly decreased after surgery, thus changing in the opposite direction than pro-inflammatory cytokines. Concentrations of TNF-α and IL6 increased at the end of the operation along with their mRNA levels in subcutaneous and epicardial adipose tissues. This finding correlates with previously published data [35]. Interestingly, other studies reported that CTRP3 is the only adipokine in septic patients that decreases significantly compared to such proteins as ghrelin, resistin, leptin, and CTRP1 [36,37,38,39]. Another study showed that metabolic comorbidities did not affect the CTRP3 dynamic in the setting of critical illness [40].

In addition to affecting glucose and lipid metabolism, CTRP3 also has anti-inflammatory properties through the inhibition of NF-κB and TLR-signaling pathways [41,42]. In our study, we demonstrate for the first time that *CTRP3* mRNA levels are differently regulated in epicardial and subcutaneous fat. Baseline *CTRP3* mRNA levels in EAT were significantly lower in +CAD+T2DM patients as compared to -CAD-T2DM patients. The same tendency that did not reach statistical significance was noted for the comparison of +CAD+T2DM patients vs. +CAD-T2DM patients. In contrast, no baseline differences in *CTRP3* mRNA levels were found in subcutaneous adipose tissue. A potential relationship of CTRP3 and cardiac function has been described previously. For example, CTRP3 attenuated pathological remodeling after acute myocardial infarction via the inhibition of myocardial fibrosis in a mouse model and improved survival and regeneration of ischemic cardiomyocytes [43,44]. Additionally, CTRP3 also improved angiogenesis through Protein kinase B phosphorylation and the expression of Hypoxia-induced factor 1-α and vascular endothelial growth factor [43]. Our results support previously suggested mechanism of CTRP3 action. It suppressed cellular apoptosis, decreased inflammatory cytokines, and opposed effects of endothelin-1 and NO by facilitating the phosphatidylinositol 3-kinase (PI3K), Akt, and eNOS productions [20]. Hence, decreased *CTRP3* mRNA expression in EAT of +CAD+T2DM group could contribute to increased local production of pro-inflammatory cytokines leading to endothelial dysfunction with subsequent promotion of coronary atherosclerosis. Our data also suggest a differential regulation of *CTRP3* mRNA expression in epicardial vs. subcutaneous fat. Due to a limited size of epicardial fat depot, its increased *CTRP3* mRNA expression did not affect total circulating CTRP3 levels which mostly come from subcutaneous adipose tissue whose *CTRP3* mRNA expression was not affected by the presence of diabetes or coronary artery disease.

Taken together, it is tempting to speculate that lower *CTRP3* mRNA levels specifically in EAT that closely adjoins myocardium and coronary arteries could contribute to the development of coronary atherosclerosis and impairment in cardiac function and angiogenesis in these patients. However, more in-depth mechanistic studies on larger patient populations are required to confirm this hypothesis.

In conclusion, our study demonstrated for the first time that *CTRP3* gene expression in EAT of patients with CAD and T2DM is significantly decreased as compared to patients without these conditions and that circulating CTRP3 levels decrease after cardiac surgery. Along with previously published experimental data, these findings may indicate a possible role of CTRP3 in the pathogenesis of coronary atherosclerosis and other cardiac pathologies.

## 4. Materials and Methods

### 4.1. Study Subjects

The study included 34 patients who underwent elective cardiac surgery with cardiopulmonary bypass (coronary artery by-pass graft). Subjects were divided into 3 groups according to the presence of CAD and T2DM: (i) 11 subjects without CAD and T2DM (-CAD-T2DM), (ii) 14 subjects with CAD and without T2DM (+CAD-T2DM), and (iii) 9 subjects with CAD and T2DM (+CAD+T2DM). Patients with T2DM were treated with metformin (3 patients), sulfonylureas (1 patient), dipeptidyl peptidase 4 inhibitors (2 patients), insulin (1 patient), or diet only (1 patient). Of participating patients, 28 had arterial hypertension and 29 dyslipidemia treated with statins. None of the patients suffered from acute or chronic kidney disease, malignancy, thyroid disease, or acute infection.

Surgery was performed after overnight fasting and was started between 07:00–08:00 in all subjects. Of the total patients, 10 received an infusion of dobutamine and norepinephrine perioperatively, with a maximum dose of 7 μg/kg/min and 0.2 μg/kg/min, respectively, with the treatment duration from 8 to 33 h.

All participants signed written informed consent prior to the enrolment into the study. The study was approved by Human Ethics Review Board, First Faculty of Medicine and General University Hospital, Prague, Czech Republic and was performed in accordance with the guidelines proposed in Declaration of Helsinki (2000) of the World Medical Association.

### 4.2. Anthropometric Examination, Blood, and Tissue Sampling

Anthropometric examination of study subjects was performed at baseline one day prior to surgery. All subjects had their height, weight, waist, and hip circumference measured, and the body mass index was calculated. Blood samples for hormonal and biochemical measurements were taken prior to initiation of anesthesia (baseline) and at the end of the operation. Serum was obtained by centrifugation and samples were subsequently stored in aliquots at −80 °C until further analysis. Samples of SAT (thoracic region, sternotomy site) and EAT (anterior interventricular sulcus or the right margin of the heart) for mRNA level analysis were taken at the start and prior to the end of the surgery from approximately the same location in all patients. Tissue samples were collected to 1 mL of RNAlater^®^ reagent (Ambion^®^—Invitrogen, Carlsbad, CA, USA) and stored at −80 °C until further analysis.

### 4.3. Hormonal and Biochemical Assays

Serum level of CTRP3 was measured using ELISA kit (Human CTRP3/C1qTNF3/CORS26 DuoSet ELISA, R&D Systems, Wiesbaden, Germany) with a limit of detection 78 pg/mL. Serum C-reactive protein (hsCRP) level was measured by high-sensitive assay (Bender Med systems, Vienna, Austria) with a sensitivity of 3 pg/mL. Serum level of TNFα and IL6 was measured by a multiplex assay MILLIPLEX MAP Human Cytokine/chemokine Panel I (Merck KGaA, Darmstadt, Germany). Sensitivity for IL6 was 0.9 pg/mL and for TNF-α 0.7 pg/mL. Serum IL6 level of most samples was undetectable by this assay, so they were repeatedly analyzed by the human IL6 high-sensitivity ELISA with analytical sensitivity of 0.03 pg/mL (Thermo Fisher Scientific, Waltham, MA, USA).

Serum leptin concentration was measured by the commercial ELISA kit (BioVendor, Brno, Czech Republic). Sensitivity was 0.2 ng/mL. Serum level of adiponectin was measured by the commercial ELISA kit (Merck KGaA, Darmstadt, Germany). The lowest level of adiponectin that can be detected by this assay is 0.2 ng/mL. The intra- and inter-assay variabilities for all assays were between 5.0% and 15.0%.

Routine biochemical parameters (Triglycerides, Total cholesterol, HDL cholesterol, Fasting glucose and HbA_1c_) were measured at the Department of Biochemistry, General University Hospital, Prague, Czech Republic by standard laboratory methods. LDL cholesterol was calculated using the Friedewald Equation.

### 4.4. Quantitative Reverse Transcription PCR

mRNA levels of *CTRP3, TNF-α* and *IL6* were determined as described in detail elsewhere [45]. Briefly, samples of tissue were homogenized on MagNA Lyser Instrument (Roche Diagnostics GmbH, Mannheim, Germany). Total RNA was extracted on MagNA Pure instrument using Magna Pure Compact RNA Isolation kit (Roche Diagnostics GmbH, Mannheim, Germany).

Reverse transcription was performed using random primers according to the manufacturer’s protocol of the High-Capacity cDNA Reverse Transcription Kits (Applied Biosystems, Foster City, CA, USA). Gene expression analysis was performed on a 7500 Real-Time PCR System using TaqMan^®^ gene Expression Assays (Applied Biosystems, Foster City, CA, USA). For the reaction, a mix of TaqMan^®^ Universal PCR Master Mix II, NO AmpErase^®^ UNG (Applied Biosystems, Foster City, CA, USA) and specific TaqMan^®^ Gene expression Assays (Applied Biosystems, Foster City, CA, USA) were used (Appendix A). Gene *Beta-2 microglobulin* was used as endogenous reference. The formula 2^−∆∆Ct^ was used to calculate relative gene expression.

### 4.5. Statistical Analysis

Statistical analysis was performed, and graphs were drawn using SigmaPlot 13.0 software (SPSS Inc., Chicago, IL, USA). The results are expressed as mean ± standard error mean (SEM) or median (interquartile range). Normality of all data was assessed by the Shapiro–Wilk test. One-way ANOVA followed by Holm–Sidak test or one-way ANOVA on Ranks followed by Dunn’s method, and a paired *t*-test or Wilcoxon Signed-Rank test were used for the assessment of intra and intergroup differences, as appropriate. Correlations were analyzed using Spearman’s or Pearson’s correlation test, according to the Normality of data. In all statistical tests, *p* values < 0.05 were considered significant.

## Figures and Tables

**Figure 1 ijms-23-09988-f001:**
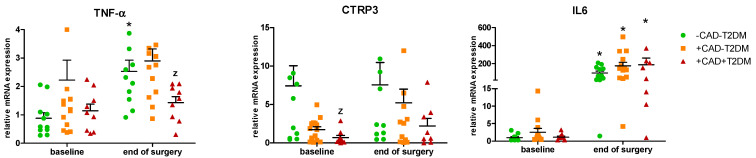
Relative mRNA quantification of *TNF-α*, *CTRP3,* and *IL6* genes in epicardial adipose tissue. Data are expressed as mean ± SEM; *p* < 0.05: ^Z^ vs. +CAD-T2DM for TNF-α and ^Z^ vs. -CAD-T2DM for CTRP3; * vs. baseline.

**Figure 2 ijms-23-09988-f002:**
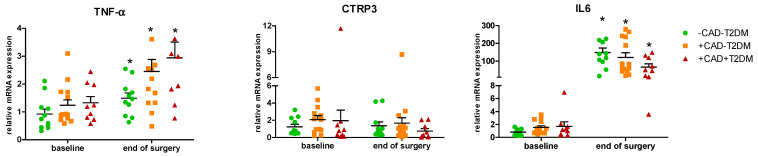
Relative mRNA quantification of *TNF-α*, *CTRP3,* and *IL6* genes in subcutaneous adipose tissue. Data are expressed as mean ± SEM; *p* < 0.05: * vs. baseline.

**Table 1 ijms-23-09988-t001:** Baseline anthropometric and biochemical characteristics of study subjects.

Parameter	-CAD-T2DM	+CAD-T2DM	+CAD+T2DM	*p*-Value
Number of subjects (f/m)	11 (3/8)	14 (2/12)	9 (2/7)	x
Age (year)	59.7 ± 4.7	69.3 ± 2.4	67.0 ± 2.9	0.132
Body mass index (kg/m^2^)	27.1 ± 1.0	28.3 ± 1.5	30.4 ± 1.2	0.252
Waist circumference (cm)	99.2 ± 3.5	103.5 ± 3.2	110.4 ± 2.7	0.089
Hip circumference (cm)	105 (100–110)	102.5 (99.5–114.8)	110 (108–115)	0.102
Waist/hip ratio	0.96 ± 0.01	0.98 ± 0.02	0.99 ± 0.02	0.521
Total cholesterol (mmol/L)	4.12 (3.40–4.48)	3.73 (2.94–4.64)	3.91 (3.69–4.60)	0.510
Triglycerides (mmol/L)	1.25 (1.00–1.45)	1.37 (0.97–1.68)	1.13 (0.98–2.08)	0.972
LDL cholesterol (mmol/L)	2.15 (1.77–2.60)	2.01 (1.46–2.71)	1.99 (1.84–2.48)	0.622
HDL cholesterol (mmol/L)	1.21 (0.95–1.62)	1.01 (0.87–1.34)	0.93 (0.86–1.81)	0.429
Fasting glucose (mmol/L)	5.4 (5.0–5.90)	5.1 (4.8–5.4)	8 (6.50–9.80) ^1,Z^	<0.001
HbA_1c_ (mmol/mol)	35.4 ± 1.2	38.1 ± 2.0	52.4 ± 3.0 ^1,Z^	<0.001

Normally distributed data are shown as mean ± SEM; non-parametric data are expressed as median (interquartile range). Statistical significance is from one-way ANOVA; ^1^
*p* < 0.05 vs. -CAD-T2DM, ^Z^ vs. +CAD-T2DM.

**Table 2 ijms-23-09988-t002:** Serum concentrations of selected cytokines and adipokines.

Analyte	-CAD-T2DM		+CAD-T2DM		+CAD+T2DM	
	Baseline	End of Surgery	Baseline	End of Surgery	Baseline	End of Surgery
hsCRP (µg/mL)	1.64 (0.94–2.92)	1.57 (0.65–2.56)	2.83 (1.25–16.02)	2.33 (0.75–13.95) *	6.04 (1.98–14.35)	5.68 (1.70–11.67)
Leptin (ng/mL)	6.19 (4.44–11.44)	3.97 (1.85–7.77) *	6.77 (3.94–9.77)	7.45 (2.93–11.76)	7.12 (5.41–16.00)	7.21 (5.66–14.38)
Adiponektin (µg/mL)	7.09 (5.21–12.83)	5.96 (4.25–7.96)	8.48 (6.12–15.13)	7.99 (6.84–12.9)	8.04 (6.93–12.01)	6.99 (5.85–9.45) *
CTRP3 (ng/mL)	60.20 ± 3.97	46.93 ± 2.88 *	64.85 ± 3.30	56.83 ± 3.67 *	60.93 ± 5.34	53.44 ± 3.99 *
IL6 (pg/mL)	0.77 (0.59–2.86)	32.43 (21.85–122.9)	2.89 (1.94–3.69)	23.98 (4.50–152.67)	3.12 (1.20–8.22)	61.87 (25.59–222.32) *
TNF-α (pg/mL)	7.61 (3.88–11.08)	9.08 (7.90–18.05) *	9.61 (6.97–13.49)	10.54 (7.09–31.74)	12.36 (6.02–20.29)	34.84 (22.37–39.53) *^,1^

Normally distributed data are shown as mean ± SEM; non-parametric data are expressed as median (interquartile range). Statistical significance is from one-way ANOVA and paired *t*-test; ^1^
*p* < 0.05 vs. -CAD-T2DM, * vs. baseline. High sensitivity C-reactive protein (hsCRP); C1qTNF-related protein 3 (CTRP3); interleukin 6 (IL6); tumor necrosis factor α (TNF-α).

## Data Availability

The data from this study are available from the corresponding author upon reasonable request.

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
