# Peer review of "Decreased Epicardial CTRP3 mRNA Levels in Patients with Type 2 Diabetes Mellitus and Coronary Artery Disease Undergoing Elective Cardiac Surgery: A Possible Association with Coronary Atherosclerosis"

_ijms, 2022, doi:10.3390/ijms23179988_

Round 1

Reviewer 1 Report

Matloch et al.'s study analyzes both circulating protein and endogenous RNA levels of adipokine CTRP3. After showing a decrease in baseline CTRP3 in EAT of CAD/T2DM study participants, they suggest that CTRP3 could be a risk factor for CAD and T2DM. However, the data as it stands does not support this and although the authors merely suggest this, they do not attempt to propose a mechanism whereby this could be further investigated.  

Circulating CTPR3 protein levels did not differ between groups and the authors show discrepancy between trends in protein expression and RNA expression (in EAT), e.g. while protein expression decreased after surgery, RNA expression did not change and while there was no difference in baseline protein expression, there was a difference in mRNA expression. Could the authors propose a mechanism for this?

While the introduction is targeted to the subject of the paper, it could benefit from some further background on CTPR3's ant-inflammatory properties and background on artherosclerosis risk factors.  

Primer details for RNA as well as raw Cq values should b e included in the supplementary.

Author Response

Dear reviewer, we thank you very much for your thoughtful and careful review and have revised the manuscript as suggested.

As per your recommendation we included more information regarding atherosclerosis and anti-inflammatory properties of studied CTRP3. We also discuss in more detail potential relationship between CTRP3 and promotion of atherosclerosis in coronary arteries.

Excel spreadsheet with Primer details for RNA used and Ct values are attached. List 1 shows details about probes used including catalogue numbers and GenBank ID. The List 2 contains Ct values and calculated 2-∆∆Ct data for EAT and in the List 3 we present 2-∆∆Ct data for SAT. More detailed information regarding probes is unavailable because we use commercial probes.

Reviewer 2 Report

This is an important topic for the coronary artery disease and type 2 diabetes. And the article present very well, and wish the authors continue to do this research, and in the future give us data in a larger sample size with another group type 2 diabetes without coronary artery disease, to confirm or improve the data today. 

As the CTRP3 could improves glucose and lipid metabolism and possesses anti-inflammatory and cardioprotective properties,the authors investigates its role in the CAD and T2D patients. And they found that, the baseline CTRP3 mRNA levels in EAT were significantly decreased in CAD/T2DM group than that in the -CAD-T2DM group. This results indicated that the decreased EAT mRNA levels of CTRP3 could contribute to higher risk of atherosclerosis in patients with CAD and T2DM. At present, the burden of cardiovascular metabolic diseases, such as diabetes and coronary heart disease, is still very heavy, and epicardial fat has become the focus of prevention and treatment of these diseases in recent years. The authors paid attention both to this major disease burden and the current hot spots, and provided new ideas for the prevention and treatment of this kind of disease. This is very good! well, some points might be improved. First, if the sample size could be larger, the results might be more confident! Second, if if there is another group,of which patients with T2D but without CAD, the results might be more interesting! Third, if the surgery kinds could be more detailed, the data might be more important!Last, If these figures are matched with different colors, the effect may be better!

Author Response

Dear reviewer, we thank you very much for your thoughtful and careful review and have revised the manuscript as suggested.

We agree with your suggestions that if the study group was larger, then the results would have been more convincing/interesting. We initially started the study on patients undergoing mechanical cardiac support implantation. However, these patients present rather as urgent cases and therefore they are deemed as a suboptimal cohort from time and logistics point of view. We could have included patients undergoing valve repairs/replacements as well, but we thought that with all patients having coronary artery bypass grafting we have at least a standardised group to avoid too many variables. And not all patients undergoing valve operation suffer from coronary artery disease.

Additionally, it would have been interesting to include -CAD/+T2DM group too, but then these patients likely would not need cardiac operation due to the unobstructive coronary atherosclerosis and therefore sampling of the epicardial fat via open sternotomy would not be justified.

We also now present some data in colours to improve quality and clarity.

To complement the revised manuscript, we state that it has been checked by a native English-speaking colleague.

Thank you very much for your time and comments.

Yours sincerely,

Martin Haluzik, MD, DSc.

Prague, August 26th, 2022